# Social Interactions in Two Groups of Zoo-Housed Adult Female Asian Elephants (*Elephas maximus*) that Differ in Relatedness

**DOI:** 10.3390/ani8080132

**Published:** 2018-08-01

**Authors:** Naomi D. Harvey, Carolyn Daly, Natasha Clark, Eleanor Ransford, Stefanie Wallace, Lisa Yon

**Affiliations:** School of Veterinary Medicine and Science, The University of Nottingham, Leicestershire, LE12 5RD, UK; Naomi.Harvey@nottingham.ac.uk (N.D.H.); svycad@nottingham.ac.uk (C.D.); svync2@nottingham.ac.uk (N.C.); svyer1@nottingham.ac.uk (E.R.); Stefanie Wallace (svysaw@nottingham.ac.uk (S.W.)

**Keywords:** elephants, zoo animals, social behaviour, affiliative, relatedness, welfare, hierachy

## Abstract

**Simple Summary:**

The chance to experience positive social interactions is important for captive animals. The amount of positive or negative social interactions that occur within groups of captive animals can be used to evaluate the welfare of the group and determine how compatible the individuals may, or may not, be. In the wild, elephants live in related, multigenerational herds that consist of mainly mothers, daughters, and their offspring. However, in captivity, they are often kept in groups of unrelated individuals, which could reduce the quality of their social interactions, and thus their welfare. Here, we recorded the social interactions between elephants in two groups of four captive female Asian elephants; in one group, all of the individuals were related to one another, whilst in the other, only two out of four individuals were related. We observed more affiliative (friendly) interactions and fewer aggressive interactions in the all-related group. We also observed elephants freely giving way to others more in the related group, with daughters giving way to their mothers and aunts, which is evidence of an established, family-based rank system that allows them to avoid escalation to aggression. These findings support the recommendation that for optimal welfare, elephants should be managed in multigenerational family herds.

**Abstract:**

Opportunities for positive social interaction are important in captive animals, and social interactions can be used as a welfare indicator. Wild elephants live in related multigenerational herds; however, in captivity they are often managed in less related groups, which could impact the quality of their social interactions, and thus their welfare. Here, we used a limited social network analysis to investigate the social interactions in two groups of four female captive Asian elephants, one of which contained individuals that were all related to one another, whilst the other was a mix of related and unrelated individuals. Data on pairwise social interactions was collected from eight days of video footage using an all-occurrence sampling technique. More affiliative, and fewer agonistic interactions were observed in the related elephant group. Additionally, non-contact displacement was observed at a higher frequency in the related elephant group, which we theorise represents an established functioning hierarchy, avoiding the need for overt aggression over resources. Although kinship is not likely to be the only factor affecting captive elephant social behaviour, these findings support the recommendation that for optimal welfare, elephants should be managed in multigenerational family herds. Evaluations of social interactions such as those conducted here would have wider applicability for aiding the management of any captive social species to identify when groups might be incompatible.

## 1. Introduction

Elephants are highly social animals that exhibit a diverse range of social behaviour [1,2,3]. Affiliative interactions observed between non-mating elephants include greeting each other by extending their trunk towards an approaching individual [4], social rubbing, and touching each other with their trunks [5,6]. Affiliative contact behaviour such as these are thought to be used for building and maintaining social bonds, providing reassurance, greeting, and play [6]. Agonistic interactions between elephants include threat displays, mock charges, biting, kicking, and other contact-based attacks that potentially inflict pain or cause another individual to move away [7]. Opportunities for multigenerational social interactions have been shown to be an important factor in captive elephants, as elephants that spend time housed alone are at higher risk of developing stereotypic (repetitive) behaviour patterns, whilst time spent with juveniles can reduce this risk [8].

In the wild, both African elephants (*Loxodonta africana*) and Asian elephants (*Elephas maximus*) form small stable groups of related adult females and their immature offspring [1,9,10]. Groups are led by a matriarch, which is usually the oldest female, who acts as a source of social knowledge [4,11]. The care of calves is central to elephant society and allomothering (when an individual other than the mother helps to care for a calf) is common [3,12]. Adult male elephants are known to live in various social situations, either travelling alone outside of their natal range, forming bachelor groups, or associating closely with female groups [1,4,10,13]. Wild African elephants whose communities have been broken apart by poaching have been documented as forming new core groups with low relatedness [14], but a higher incidence of agonistic interactions is observed within these less-related core groups than within highly related groups with old matriarchs. Much of what is known about elephant behaviour has come from studies of wild African elephants, and some aspects of this may not be applicable to Asian elephants [2,15].

Social behaviour has not been as extensively studied in the wild Asian elephant, possibly because their dense forest habitat makes it more difficult to conduct observations [6,9,16]. However, it has been established that core family groups are formed that are comparable to the multiple mother–calf matriarchal units seen in African elephants [9,10,17]. Wild Asian elephants in Sri Lanka that associate with each other have been shown to have a shared maternal ancestor, indicating a connection between relatedness and social association [9]. No study of wild Asian elephants has been able to find evidence of adult female intergroup transfers, suggesting that females remain in their natal groupings [9,10].

A 20-month observational study of 286 wild Asian elephants in Sri Lanka concluded that the average core group size consists of two to three individuals, with 18% of females forming much stronger relationships to one individual, relative to all other social ties [16]. Overall, the group dynamics observed consisted of a few core consistent relationships surrounded by a fluid fission–fusion network of changing companionships, with weaker social ties than are generally seen in African elephants [16]. Although elephant societies are usually characterised by fission–fusion dynamics, evidence from wild African elephants shows that when a group separates, the adult females remain with their first degree relatives, and that when groups fuse temporarily, the oldest females in each group tend to be related [18]. Close associations between non-related individuals have been observed when elephants are raised together when populations are disturbed [19,20], as well as in captive adult female Asian elephants [21], but these likely act as substitute bonds in the absence of matrilineal kin [19], and may not be of equal quality to familial bonds.

The well-being and quality of life of captive elephants has become a growing concern [22], and the Elephant Welfare Group (EWG) was set up in the United Kingdom (UK) to provide evidence on the welfare status of UK captive elephants [23,24]. A welfare assessment tool was developed by the EWG for use in monitoring the welfare of captive elephants, of which social behaviour formed an important part [25]. Zoo elephants are often housed with unrelated individuals [26], which is different to the social structure of their wild counterparts [9,10]. There are 39 Asian elephants kept at eight different facilities in the UK and Ireland [27]. The British and Irish Association of Zoos and Aquariums (BIAZA) recommended that elephants are housed in groups of at least four cows over the age of two years. According to these guidelines, the group should be socially compatible, and if possible, individuals should be related to one another to emulate wild group composition [28].

The few studies of captive Asian elephant social behaviour that collected data on related and unrelated individuals have identified a higher frequency of affiliative interactions and social associations between related individuals compared with unrelated individuals [29,30,31,32], with the strongest social bonds being between calves, and calves and their mothers. However, these were small in size, focussing on just one group of animals at any one time, and further evaluation of captive elephant social dynamics is warranted. It has been theorised that agonistic behaviour and tension will be lower in elephant groups who are more highly related [33]. If enough evidence can be gathered to support this, such a finding would have direct implications for the management of captive animals to maximise their welfare.

The aim of this study was to confirm the impact of group composition, in terms of relatedness, on affiliative and agonistic interactions by comparing the social network structure of two groups of captive elephants, which differed in overall relatedness. Based upon previous work, we hypothesised that there would be a higher frequency of affiliative interactions, and fewer agonistic interactions, in the more related group, and stronger affiliative ties between members of the more related group compared with the less related group.

## 2. Materials and Methods

### 2.1. Subjects

This study was conducted at two zoos that housed Asian elephants within the UK and Ireland. The study subjects consisted of eight adult female Asian elephants: four at each zoo (Table 1). From here on, each elephant will be referred to by their individual ID.

At Zoo A, only two of the study subjects were related, A1 and A4, who were mother and daughter (Appendix A). The other subjects were not related, although the group was housed with A3’s calf, who was not included in this study. At Zoo B, all of the study subjects were genetically related (Appendix A). B1 and B2 were full sisters, and B3 and B4 were their daughters, meaning that B1 and B2 were aunts to B4 and B3 also. The elephant group at Zoo B was housed with the calves of B1, B2, and B3, who were not included in this study. At Zoo B, a bull elephant was housed with the study subjects between the hours of 10:00 and 15:30, but was not included in this study, as these hours did not span the entire collection period.

### 2.2. Elephant Management

#### 2.2.1. Zoo A

The elephant habitat at Zoo A consisted of an indoor house, a sand paddock, and a grass paddock (Appendix A). The sand paddock, of approximately 2491 m^2^, contained boulders, tree trunks and branches, raised ground, and a pool. The adjacent grass paddock was approximately 3621 m^2^. The indoor house consisted of concrete flooring. Between 10:00 and 16:30, the elephants only had access to the outside areas, although access was given to the indoor house if the weather was unsuitable. At other times, access to the entire enclosure was given, except if keepers were cleaning an area of the enclosure. Elephants were given a main feed just before 10:00, when they were given access to only the outside areas. At various points in the day, elephants were provided with browse, in the form of branches, at the edge of the sand paddock. At 14:30 every day, keepers gave a public talk and provided food and browse. An evening main feed was given just before they were given access to the inside house at 16:30. Further browse, in the form of hay, was provided in the elephant house.

#### 2.2.2. Zoo B

The elephant habitat at Zoo B consisted of an inside house, a cow paddock, and a bull paddock. The inside area covered 320 m^2^, the cow paddock covered 7000 m^2^, and the bull paddock covered 3000 m^2^. Within the paddocks there were boulders, trees, raised ground, and two pools. Hay nets were provided on crane structures. The indoor house substrate consisted of deep sand that was shaped into mounds for elephants to rest against at night. Inside the house, another two hay nets were hung, and automatic timer feeders were provided on two walls. Between 10:00 and 15:30, elephants only had access to the outdoor sand paddocks if the weather was suitable, during which time the bull was included with the group. At 15:30, the bull was separated into the bull paddock, and the cows were given access to both the inside house and cow paddock. The main feed was given during training sessions between 09:00–10:00 every day. At 15:30, the cows had access to further browse in the form of branches and hay in the indoor house, and various vegetables and fruits from four feeder boxes provided around the edge of the house. The automatic feeders also triggered at various times overnight.

#### 2.2.3. Piloting

A draft social interaction ethogram was adapted from the relevant sections of an existing ethogram developed by the Elephant Welfare Group (Appendix A). This draft ethogram was tested by Carolyn Daly during three days of live observations at both Zoo A and Zoo B, during which time pilot data was collected through continuous recording, in which all of the behaviours were recorded as they occurred in a given sampling period. On the first day at each zoo, Carolyn Daly learned to identify each individual elephant, became familiar with their husbandry routine, and conducted some preliminary observations. Live observations were then conducted for three and half days using the draft social behaviour ethogram. All of the observations took place from the public viewing areas of the enclosure. A focal elephant was selected and observed for social behaviours for a specified amount of time (a focal sampling interval) within a 1.5-h sampling period.

During preliminary observations at Zoo A, focal sampling intervals of 5 min, 10 min, 15 min, and 20 min were trialed. An interval of 15 min was selected, as the shorter intervals were less practical as the observer had to move around more often to find each elephant. Observations on the focal elephant were paused when the elephant was out of view or keepers were affecting their behaviour in some way, and resumed when the elephant was back in sight and their behaviour was not being affected. Once one elephant had a total of 15 min of focal observations, the next elephant was observed until all of the individuals had 15 min of observations in each sampling period.

At Zoo B, it quickly became apparent that live observations would not be feasible. There was a high frequency of interactions between the elephants; often, the focal elephant interacted with multiple elephants at once, making social interactions difficult to record with accuracy. Further, there were areas of the elephant enclosure that were not visible from the public viewing areas, resulting in many pauses in data collection. At some points, the focal elephant could not be located within the enclosure at all. Consequently, it was decided by Carolyn Daly, Naomi D. Harvey, and Lisa Yon that full data collection would be conducted using video footage from each zoo.

#### 2.2.4. Behavioural Observations—Full Study

The draft ethogram was refined following piloting (Table 2). The score ‘approach’ was removed, as it was ambiguous and hard to identify if an elephant was just passing another, or actually making an approach towards it. ‘Pushing’ and ‘nudging’ were observed to also cause displacement at times, so the category ‘displacement’ was broken into ‘contact displacement’ and ‘non-contact displacement’, with only contact-displacement presumed to be agonistic. No assumptions were made about non-contact displacement, which was treated separately to affiliative & agonistic interactions for analysis. ‘Tail pulling’ was added as an additional agonistic interaction, and an ‘other’ option was added for social behaviour to be recorded that was not already covered in the ethogram. The initiator and receiver of each interaction was recorded.

Only the inside video footage at Zoo B was found to have a high enough resolution to enable individual elephants and subtle interactions to be identified. Consequently, data was only collected from the inside houses at both zoos in order for the data to be comparable. Elephants were found to be most visible, at both zoos, during the early morning before keepers arrived, when they were let into the inside house to access food in the afternoon and throughout the evening. Therefore, sampling fames were limited to these time points in both zoos: early morning 06:00–08:30; inside feeding session 15:30/16:30 to 17:30/18:30; evening session 18:30–21:00. At both zoos, footage was recorded within the elephant house using the zoos’ onsite cameras, which covered the entire house from four different views at Zoo A and three different views at Zoo B.

In order to maximise the amount of data that could be collected, all-occurrence sampling was used, in which all of the social interactions between all of the study subjects who were in view during the sampling frames were recorded [34]. Data recording was paused when keepers were present or there were no elephants in view of the cameras, and resumed when there were at least two elephants in view of the cameras and keepers were not present. Recording continued until 1 h of active recording had been conducted within each sampling frame, so that the period of observations was balanced between zoos.

Video footage was collected from each zoo during two time periods in the spring and autumn of 2015. At Zoo A, four days of spring footage were collected between 27 and 30 April, and four days of autumn footage (spread over five calendar days due to damaged files) between the afternoon of 7 September to the afternoon of 11 September.

Data was recorded directly into a Microsoft Excel spreadsheet (version Office 2013) for each sampling period. The datasheet detailed the sampling start and stop times, elephants in view, the initiator and receiver of each interaction, and the option to note and describe “other” behaviour.

#### 2.2.5. Elephant Identification and Observer Training

Video footage was analysed by Carolyn Daly, Eleanor Ransford, Stefanie Wallace, and Natasha Clark. Individual elephants were identified through physiological features including their height, tail length, back conformation, presence of hair on the tail or back, presence of ‘tushes’ (small proto-tusks), characteristic ear shape and folds, and other prominent features recorded in a photographic identification catalogue. Additionally, Carolyn Daly, Eleanor Ransford, Stefanie Wallace, and Natasha Clark each visited both zoos in person before they began scoring videos to observe the elephants live in order to familiarise themselves with the individuals. Before beginning data extraction, each observer watched and coded the same day of footage independently, and then checked their data against each other’s. Where disagreements occurred, the footage was re-watched as a group until consensus was reached on the behaviour observed and/or individual identification. This process was repeated twice more until all of the observers were identifying the same individuals and behaviour.

#### 2.2.6. Social Behaviour Analysis

Data was analysed using a limited social network analysis (SNA), as only a limited set of SNA features could be applied to the small group sizes (*n* = 4) in this study. Using SNA, we represent the relationships between individuals (nodes) and can take into account the effect that a single node may have on other nodes or the population (see Table 3 for term definitions). Each node in the group may have a social “tie” to other nodes. A social network is formed by a collection of social ties between nodes within a group [35,36]. Relationships studied in a social network can encompass a number of different interactions such as agonistic or affiliative interactions. Agonistic interaction networks could be used to describe a group’s dominance hierarchy, whilst affiliative networks could give some insight into the stability of a group [36].

We recorded social interactions for a total of 11.85 h from the autumn video footage (8.9 min of data was missing at Zoo A due to file damage, so Zoo B’s data was restricted in the equivalent time period so as to be directly comparable), and 12 h from the spring footage.

Behaviour recorded as “other” was classified as affiliative or agonistic by the entire research team. Contact displacement was included in the agonistic analyses, and non-contact displacement was considered to be ambiguous as to whether it was agonistic or simply a function of a healthy dominance hierarchy. Therefore, analysis was carried out on the affiliative, agonistic, and non-contact displacement data separately.

As a measure of individual sociality, the network metric *strength* was recorded for each type of interaction for each individual by summing the number of interactions that each individual had with all of the others in the group [37]. Video footage did not cover the entire area of the elephant enclosures, so some elephants had more time in view than others. To ensure that all of the individuals were equally represented in the data, *strength* scores were adjusted as per the recommendations in Farine and Whitehead (2015) [38]. The minimum number of hours in view for each elephant in each sample period was found to be 5 h, so the *strength* for each individual was adjusted to represent the rate of interactions observed in a 5-h period as per the following equation:$$5\mathrm{hrstrength}=\left(\frac{\mathrm{rawstrength}}{\mathrm{totalhoursinview}}\right)*5$$

Consistency of individual behavioural sociality over time (a form of test–retest reliability) was determined by comparing the rank order of *strength* per 5 h between the spring and autumn samples using Spearman’s correlation coefficients [37].

For each dyad, each type of interaction (affiliative, agonistic, and non-contact displacement) was summed to give an initiating and receiving frequency for each node, and then adjusted to represent 5 h of observation time. These values were then summed between the two sampling points (thus representing 10 h of observation per individual) and used to construct an asymmetric matrix for each type of interaction per zoo (Appendix A).

Weighted digraphs were constructed from each asymmetric matrix for each type of interaction [35]. A social tie was considered to exist between two nodes if a social interaction had occurred. The direction of the interaction, from the initiator to the receiving elephant, was shown using arrows. The frequency of the interaction determined the weight of the tie (thickness of arrow). A standardised arrow weight scaling was used between zoos and digraphs, so the results would be directly comparable.

## 3. Results

### 3.1. Consistency over Time

Statistically significant Spearman’s rank correlations of 0.85 (*p* < 0.001) were evident in adjusted *strength* scores for all of the social interactions between time 1 and time 2, suggesting a strong consistency in individual sociality over time, supporting the aggregation of the spring and autumn data (Figure 1 and Appendix A).

### 3.2. Behaviour Frequency

When comparing the sum of adjusted *strength* scores between zoos, the elephants in Zoo B (the related group) exhibited 41% more affiliative interactions, 229% more non-contact displacement, and 63% fewer agonistic interactions than the elephants at Zoo A (Figure 2).

With regard to the specific types of affiliative interactions, no elephants in either zoo were observed feeding each other or engaging in social play. The most commonly observed behaviours were trunk touching, rubbing/nudging another, and leaning, all of which were observed more in Zoo B (Figure 3). ‘Other’ affiliative behaviour that was observed only at Zoo B included three instances of B4 holding her aunt’s (B2’s) tail.

A limited range of agonistic behaviour was observed at both zoos, with six of the 15 defined in the ethogram observed at Zoo A, plus one additional ‘other’ behaviour, and four of 15 types observed at Zoo B, plus three additional ‘other’ behaviours (Figure 4). Biting, lunging, and tail pulling were only observed at Zoo A, with all of the instances of tail pulling and lunging being initiated by A3 towards A1. All but one case of ‘other’ agonistic behaviour that was observed at Zoo A consisted of repetitive head bumps directed from A3 towards A1, and one instance of A1 physically blocking A3’s access to a gate. At Zoo B, the ‘other’ category consisted of five instances of food stealing, and 10 instances of blocking access to food by swatting away another’s trunk (four) or physically blocking food access with their body (six).

### 3.3. Affiliative Network

With regards to the affiliative network, the most balanced dyad in Zoo A was the mother–daughter pair; all of the others were imbalanced, with one individual directing more affiliation than they received within a dyad (Figure 5). A1 received the most affiliative interactions from all of the members of the social group, suggesting that A1 was central to the group dynamic.

In Zoo B, the most balanced dyad was the cousins (B4 and B3). Considerably more affiliative interactions were observed in Zoo B, primarily by the younger members, who initiated more affiliative interactions than did their mothers. B4 initiated the most affiliative interactions overall, with all of the members of the group.

### 3.4. Agonistic Network

In Zoo A, A3 was responsible for the majority of agonistic interactions; these were directed mostly towards A1 (Figure 6). A1 also received the most affiliative interactions from A3. When looking at the raw data, the nature of A3’s relationship with A1 appeared to change between the spring and autumn (a breakdown of raw interactions directed towards A1 from A3 can be seen in Appendix A). In the spring, A3 directed 32 affiliative interactions and 17 agonistic interactions towards A1, whilst in the autumn, the inverse was seen, with A3 directing just 10 affiliative interactions and 28 agonistic ones towards A1. When considering interactions directed from A1 towards A3, A1 interacted very little with A3 at any time point, totaling just eight interactions directed at A3 in the spring (five affiliative/two agonistic/one non-contact displacement, or NCD), and zero in the autumn of any kind.

For Zoo A, the fewest agonistic interactions were seen between the mother–daughter dyad (just one from mother to daughter), and A4 did not initiate any agonistic interactions in either observation period. In Zoo B, no strong agonistic relationships existed between any dyads, with the most seen being eight incidences directed from B2 to her sibling, B1.

### 3.5. Non-Contact Displacement Network

Very little NCD was observed amongst the elephants at Zoo A (Figure 7). A2 displaced all of the other group members at least once, but was never displaced herself, whilst A4 displaced no group members, but was the most often displaced individual, mainly by unrelated members A1 and A2.

Considerably more NCD was observed in Zoo B; these occurred more than agonistic interactions, which was the opposite to Zoo A. In Zoo B, there appeared to be a hierarchy of displacement, with B2 displacing her sibling B1 most often, followed secondarily by her niece B4, whilst B1 primarily displaced her niece B3, and secondarily displaced her daughter B4.

On no occasion in either zoo was a daughter observed displacing a mother; additionally, at Zoo B, no nieces were observed displacing their aunts.

## 4. Discussion

When it comes to animal welfare, especially for social species, an important but often overlooked aspect of welfare is the compatibility of the social group. Employing social network analysis can provide valuable insight into the group dynamics of captive animal groups, which can aid keepers in their management [29,39].

Despite the fluid, fission–fusion nature of wild elephant societies, related female matrilines form the core of each group, with first-degree maternal relatives remaining together when groups split up [18]. Due to this, Veasy [33] theorised that agonistic behaviour would be lower in more highly related captive elephant groups. Here, we observed significantly lower levels of agonistic behaviour and greater levels of affiliative behaviour in the related group compared with the less related group, supporting Veasy’s hypothesis.

In addition to agonistic and affiliative behaviour, we observed a form of deferential displacement of one individual by another, which we named non-contact displacement, but is often referred to as ‘supplanting’ [34]. This non-contact displacement exhibited significant interindividual consistency over time, occurred at a much greater frequency in the related group of elephants, and was predominantly initiated by the older females. Whilst dominance hierarchies may be founded by agonistic interactions, they are maintained by one individual in a dyad consistently yielding to another to avoid escalation [40]. Due to the lower frequency of agonistic interactions and higher frequency of non-contact displacement in the related group, we hypothesised that this behaviour was evidence of a functional hierarchy in the related group that was lacking amongst the less related group.

Animal behaviour researchers are often interested in evaluating whether a dominance hierarchy exists amongst groups of animals (e.g., [29,41]). However, the usefulness of the concept has been questioned [40,42,43,44]. The concept of dominance is broad, and there are various different functional and descriptive definitions [40]. No matter the definition, dominance hierarchies exist only within the group studied; they will change as the group changes, and may only be reflective of a singular context. For example, there may be a different hierarchy for access to food compared with access to water. In the wild, female Asian elephants have well-resolved hierarchies that are ordered by age/size, where older larger females dominate smaller/younger ones for access to resources [45]. However, caution must be urged when considering dominance hierarchies in the case of captive animals in small groups, as in groups of six or less individuals, there is a high probability that hierarchical linearity will occur by chance [34,46]. In the case of captive elephants, we suggest that understanding the frequency of agonistic and affiliative behaviour in addition to non-contact displacement between individuals within elephant groups would be more beneficial than focusing on identifying dominance hierarchies, per se.

When group composition changes due to new additions/subtractions, it is possible that this will lead to the need to establish one or more new dominance hierarchies, which could be founded by agonistic interactions. However, if these agonistic interactions are maintained at a higher frequency than deferential non-contact displacement, as we saw here in Zoo A, it may be a sign that a group is incompatible. Thus, we propose that it is crucial to evaluate and compare the frequency of supplanting in studies of group dynamics, relative to the frequency of agonistic and affiliative behaviour. This issue of the dynamics between different types of interactions deserves further exploration in future studies, particularly for the evaluation of captive animal welfare.

As with captive Asian elephants at Knowsley Safari Park [29], the affiliative ties seen here were most often unbalanced, with most individuals not equally reciprocating the affiliative interactions that they received, and the most affiliative behaviour across all of the individuals was exhibited by the youngest members of the study (B3 and B4, who were aged eight and 12). It could be argued that the higher rate of affiliative interactions observed amongst the related group is due to the younger age of B3 and B4 at Zoo B. However, when our data were combined with published data on the rate of interactions seen between female Asian elephants at two other zoos [29,32], the average hourly rate of affiliative interactions amongst 36 unrelated dyads was 0.98, and amongst 11 related dyads was 2.80 (Appendix A). When interactions involving juveniles were removed, the average rate for related adult individuals was still almost double that for unrelated individuals at 1.81, compared to 1.05. This lends strength to the supposition that the increased rate of affiliative interactions observed at Zoo B was at least partly due to the increased relatedness of the group.

The unbalanced nature of the social ties observed in captive Asian elephants could be a reflection of the fission–fusion nature of wild Asian elephant social dynamics, where most ties are weak, with only 18% of females tending to form strong ties [16]. It has been suggested that species with fission–fusion networks may cope better under captive management regimes [47]. However, the main difference between the wild and captivity is the lack of choice that an individual has regarding who is available for them to associate with. This can be exacerbated when incompatible individuals are kept in small groups with limited space to disperse. Such a lack of choice may negate the ability for an animal to exhibit social flexibility. In a study of captive African elephants without spatial constraints, the elephants chose to distance themselves by at least five body lengths, and exhibited few social interactions, of which most were affiliative [48]. Whilst such research suggests that dispersing access to resources and larger enclosure sizes could reduce aggression in captive elephants, if they are unrelated they may still not have the opportunity to engage in positive familial social interactions, and the chance to experience positive social interactions is important for an animals’ quality of life. Here, A1 was observed directing 36 agonistic interactions towards A3; this is 12 times the median frequency of agonistic interactions observed across both zoos, and 2.7 times higher than the second largest agonistic tie in either zoo. When combined with other published data [29,32], the mean hourly rate of agonistic interactions seen between 36 dyads of unrelated female Asian elephants was 0.36 interactions per hour, whilst for A1 and A3, the hourly rate was 3.7, which was the highest of all 36 dyads (Appendix A). The average rate of agonistic interactions in the related dyads from this comparison was 0.25, which was lower than for unrelated individuals. High levels of agonistic interactions directed at specific individuals, as we observed here directed towards A1 by A3, suggest that the two individuals were not compatible. Such interactions are likely to cause chronic low-level stress, which would be a welfare concern for the animals involved. This highlights the need to monitor social relationships amongst captive animals to identify individuals that may not be compatible and may be suffering compromised welfare.

If used longitudinally to track changes in the social dynamics of captive living animals, social network measures could prove to be valuable predictors of the factors involved in group cohesion, allowing for improvements to be made in predicting social compatibility (e.g., [39]). Given that the BIAZA guidelines for managing captive elephants place an emphasis on the need for the group to be socially compatible, regular recording of social network measures in captive elephant groups would be recommended. If this could be conducted in a comparable manner across institutions, it would enable greater insight into the factors affecting social cohesion in captive elephants, such as species, group size, sex composition, origin (wild born or captive born), length of time together, seasonal variations, and resource distribution. Such insight could be utilised to improve captive elephant management internationally.

### Study Limitations

Due to limitations in video quality, observations could only be made of interactions that occurred inside the elephant houses. It is possible that social dynamics could have been different outside of the elephant houses due to a variety of different factors, such as food access, the weather, enclosure design, and access to the bull at Zoo B. The data was also not representative of a full 24-h period, as data was collected at three different points during daylight hours. Further, as data was collected indoors and around food, proximity could not be used as an indicator of relationship strength, which has been shown to be a good indicator of the presence of affiliative social relationships in elephants [21,49]. This limitation meaning that only overt displays of social interaction could use utilised as measures of affiliative relationship strength.

As a function of the small group sizes at each zoo, only a limited set of social network analysis features could be utilised. Features such as ‘density’, which would give an indication of a group’s cohesiveness, could not be calculated. As with other studies, the sample size here was small, containing elephants from only two zoos. However, when integrated with other existing data, our results lend support to recommendations that relatedness is important when considering the welfare of captive elephants.

The aim of this study was to compare social interactions between a related and unrelated group of captive elephants, and so this has in turn shaped our conclusions. Therefore, it is possible that the differences that we have reported here were influenced by factors other than group relatedness. Both zoos used protective contact to manage their elephants; the elephants were the same species and same sex, but differences in personality, keeper–animal relationships, the past history of the individuals, enclosure types, and enrichment schedules could all play a part in group cohesiveness.

## 5. Conclusions

We compared the frequency and type of social interactions exhibited by two groups of female Asian elephants at two different zoos to test the hypothesis that less agonistic behaviour would be observed in the group comprised of all related matrilineal females. As hypothesised, lower levels of agonistic behaviour and greater levels of affiliative behaviour were observed in the related elephant group. Further, the related group exhibited more non-contact displacement than they did agonistic behaviour, whilst the inverse was true for the less related group. We conclude that the balance between the frequency of agonistic behaviour and deferential non-contact displacement is of crucial importance in determining whether an elephant group is compatible. Where a greater frequency of agonistic behaviour than non-contact displacement is observed and sustained in a captive elephant group, it implies that at least some individuals in the group may not be compatible, and steps should be taken to rectify the situation for the well-being of the individuals involved. Regular recording of social behaviour between dyads within captive elephant groups would provide valuable insight into group dynamics that could aid keepers in the management of their animals. These findings are applicable not only to elephants, but to any social species kept in a zoo environment.

## Figures and Tables

**Figure 1 animals-08-00132-f001:**
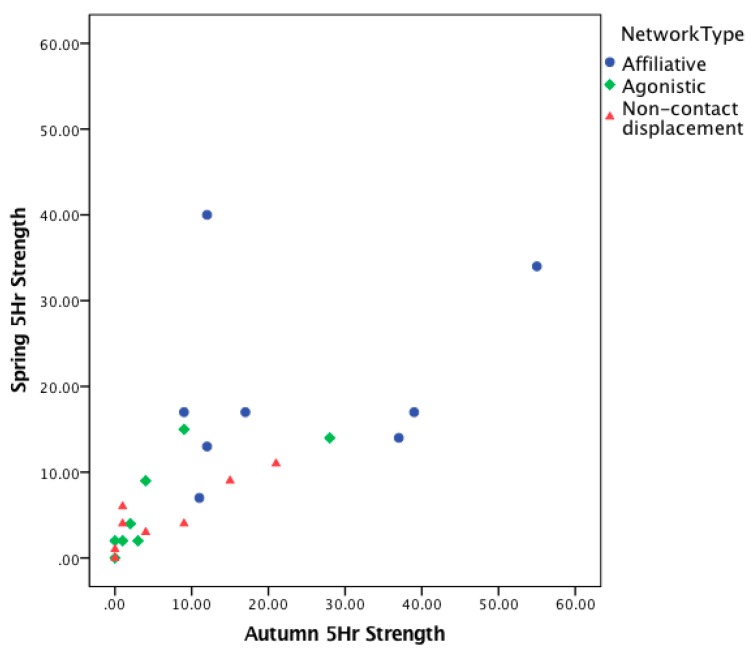
Individual consistency in social behaviour over time. Scatter plot showing adjusted individual elephant’s (*n* = 8) *strength* scores for three different classifications of social interaction (affiliative, agonistic, and non-contact displacement) between two time points (autumn and spring). Strength scores control for differences in amount of time that individuals were in view of the cameras, and represent 5 h of observation for each individual in each sample period.

**Figure 2 animals-08-00132-f002:**
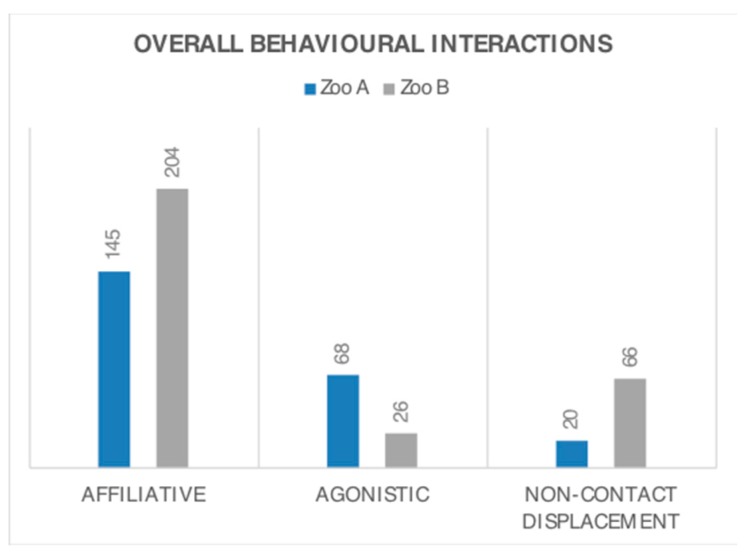
Comparison of overall interaction counts between zoos. Sum of interactions (adjusted strength) per social interaction category observed in each zoo across both time points, representing 10 h of observation per individual (*n* = 4 per zoo).

**Figure 3 animals-08-00132-f003:**
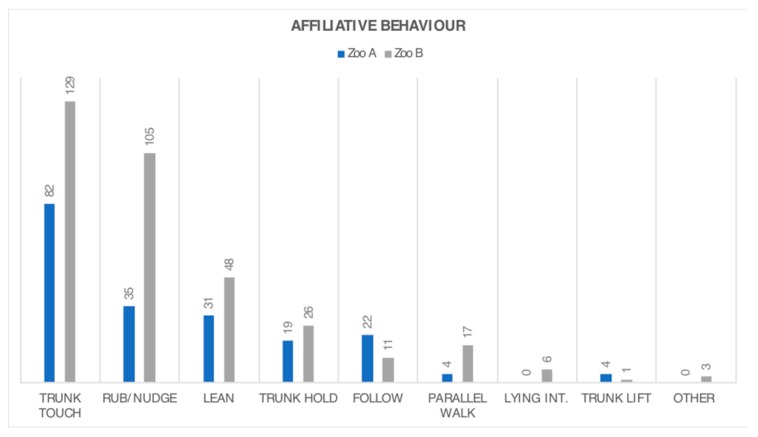
Frequency of observed affiliative interactions between the female elephants at Zoo A and B. Data from combined spring and autumn observations are given in terms of raw frequencies.

**Figure 4 animals-08-00132-f004:**
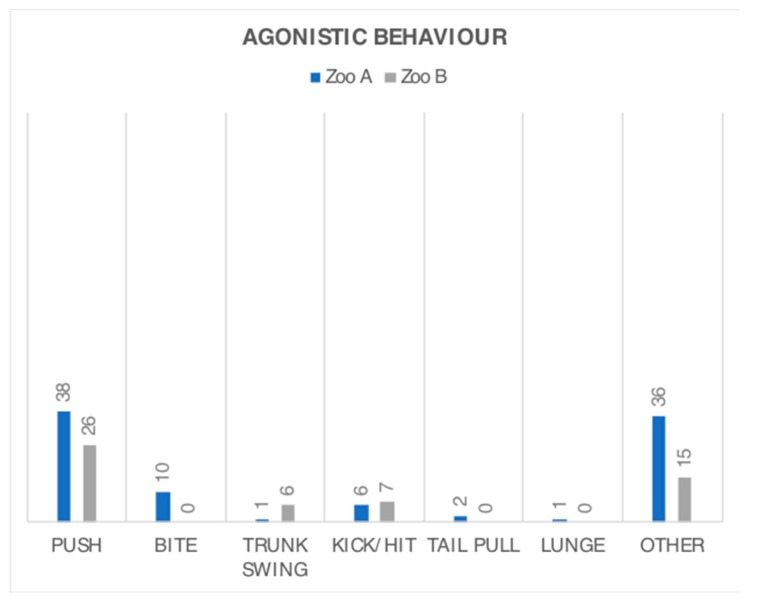
Frequency of observed agonistic interactions between the female elephants at Zoo A and B. Data from combined spring and autumn observations are given in terms of raw frequencies and presented on the same scale as Figure 3.

**Figure 5 animals-08-00132-f005:**
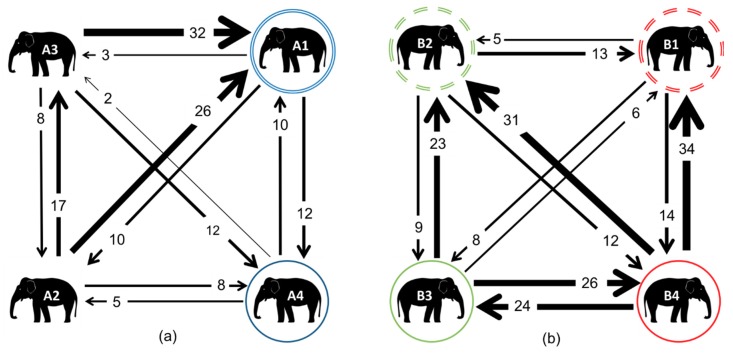
Weighted digraphs based upon affiliative interactions for the elephants at Zoo A (**a**) and Zoo B (**b**). The double circles indicate mothers, whilst matching coloured solid circles indicate their daughters. In Zoo B, all of the individuals were related: dashed circles indicate siblings; their daughters B3 and B4 are therefore cousins, and nieces to the other sibling.

**Figure 6 animals-08-00132-f006:**
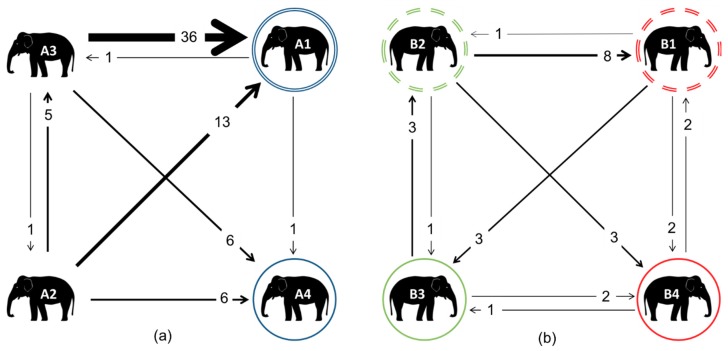
Weighted digraphs based upon agonistic interactions for the elephants at Zoo A (**a**) and Zoo B (**b**). The double circles indicate mothers, whilst matching coloured solid circles indicate their daughters. In Zoo B, all of the individuals were related: dashed circles indicate siblings; their daughters B3 and B4 are therefore cousins, and nieces to the other sibling.

**Figure 7 animals-08-00132-f007:**
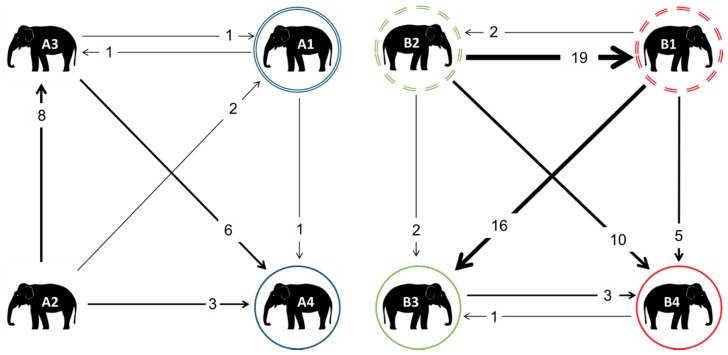
Weighted digraphs based upon the non-contact displacement for the elephants at Zoo A (**a**) and Zoo B (**b**). The double circles indicate mothers, whilst matching coloured solid circles indicate their daughters. In Zoo B, all of the individuals were related: dashed circles indicate siblings; their daughters B3 and B4 are therefore cousins, and nieces to the other sibling.

**Table 1 animals-08-00132-t001:** Subjects demographics. Elephants are listed individually with origin and age at the time of the study.

Zoo	Individual ID	Origin	Age
A	A1	Wild caught	31 (est)
A	A2	Wild caught	31(est)
A	A3	Captive bred	19
A	A4	Captive bred	17
B	B1	Captive bred	31
B	B2	Captive bred	24
B	B3	Captive bred	12
B	B4	Captive bred	8

**Table 2 animals-08-00132-t002:** Descriptions of social interactions used in the ethogram for data collection.

Type	Behaviour	Descriptions
Affiliative	Leaning	Leaning on another elephant
Affiliative	Lying interaction	Standing over another elephant—usually one that is lying down or young; Placing at least one foot on top of another elephant—usually one that is lying down; Sitting in a crouched position on top of another elephant that is in lying rest
Affiliative	Trunk touch	Putting the trunk in the mouth of another elephant; Touching another elephant, not the mouth or genitals, with the trunk in a non-aggressive manner; Touching the genital area of another elephant with the trunk;
Affiliative	Trunk hold	Intertwining of trunks between two elephants
Affiliative	Body rub or Nudge	Gentle physical contact between two elephants, which may be head–head, head–body, or body–body (not including touching with trunk); Rubbing the body against another elephant
Affiliative	Parallel walk	Two elephants walking side by side in a non-aggressive manner, for three or more steps
Affiliative	Follow	One elephant walks closely behind (within two elephant body lengths) of another elephant
Affiliative	Offer food	One elephant pushes a pile of food towards another elephant, looks like an offering of the resource
Affiliative	Trunk Lift	Trunk is outstretched and raised towards an approaching individual
Affiliative	Play	Engaging in active play with another elephant, including head-to-head sparring, trunk wrestling, mounting, chasing, and rolling on one another. Does not include behaviour observed following an antagonistic encounter or as part of courtship
Agonistic	Kick or hit	Strike out or hit an elephant with a foot in a seemingly aggressive manner—may include kicking of sand towards another elephant; Hitting another elephant with the trunk or tail
Agonistic	Charge	Move towards another elephant with the head held high, pace usually quickens as individual gets closer to the target elephant
Agonistic	Chase	Charge leading to pursuit of another elephant
Agonistic	Push	One elephant forces or pushes against the body (usually the rump) of another elephant, resulting in the elephant that is being pushed moving at least two steps
Agonistic	Stand off	Two elephants standing facing in opposite directions with foreheads pushing against each other
Agonistic	Floor smack	Hitting the trunk on the floor in an aggressive manner, may be accompanied by a ‘snort’
Agonistic	Lunge	A lunging motion followed by physical contact, used to prevent another elephant standing up
Agonistic	Tusking	Poking or jabbing at another elephant with the tusk
Agonistic	Tail Pulling	Sharply pulling the tail of another elephant
Agonistic	Directed trunk swing	Head oriented towards another elephant, human, or change in the environment, violently swinging the trunk around in an aggressive display
Agonistic	Contact displacement	Movement of one elephant resulting in another elephant leaving its location (within 10 s) caused by physical contact between individuals such as a push or nudge
Agonistic	Aggressive display: standing	Facing another elephant in an aggressive posture; Head held high, Ears wide or flapping
Agonistic	Aggressive display: walking	Facing another elephant while walking with head bobbing up and down or side-to-side, ears wide or flapping
Agonistic	Size up	Two elephants directly facing each other, standing as tall as possible, heads raised, and ears spread wide
Agonistic	Bite	Biting of the body, trunk or tail of another elephant
Non-contact displacement	NCD	Movement of one elephant towards the other, resulting in another elephant leaving its location (within 10 s)—no physical contact occurs between elephants
	Other	Describe

**Table 3 animals-08-00132-t003:** Definitions of the terms used in the social network analysis based upon Coleing [29], Wasserman & Faust [35], and Wey et al. [36].

Term	Definition
Dyad	A pair of individuals
Node	Represents an individual or study subject
Tie	Represents the social interactions between two nodes
Network	A collection of ties between nodes
Asymmetric matrix	A grid containing each node along the horizontal and vertical axes. The horizontal axes represent the initiating elephant, and the vertical axes represent the receiving elephant. The frequency of an interaction from the initiator to the receiver is recorded in the intersecting square for that dyad.
Digraph	A graph in which nodes are connected by social ties. An arrow is used to show who initiated and received the social interaction within the dyad.
Weighted digraph	A graph in which the arrows are weighted with the frequency or strength of the social interaction. Each weighted tie has a direction to indicate who initiated and received the social interaction within a dyad.

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
