# Peer review of "Social Interactions in Two Groups of Zoo-Housed Adult Female Asian Elephants (Elephas maximus) that Differ in Relatedness"

_animals, 2018, doi:10.3390/ani8080132_

Round 1

Reviewer 1 Report

Comments:

I do think this manuscript is well written and that this study is a necessary study. The following are my comments on the manuscript:

In the Introduction section, you mention in the first paragraph that, "Agonistic interactions between elephants include threat displays, mock charges, biting, kicking, and other contact-based attacks that inflict pain . . ." My concern is how was it determined that these behaviors inflict pain? 

There are a couple of citations missing, like Makecha et al. (2012), which concerns social relationships in captivity, and Garai (1992), which concerns special relationships in elephants (you  mention in the 3rd paragraph that it is not known if unrelated elephants brought together as adults in captivity can form such bonds. 

There are more captive studies on elephant social behavior than what is cited in the Introduction section (top of 4th paragraph), so I disagree with the statement starting with, "The few existing studies of captive Asian elephant social behavior . . . "

There are some spelling errors (not many) throughout the manuscript which can be easily addressed. One of the recurring error is using "zoo's" vs. "zoos"

One of the things that I would like to see clarified are how affiliative and agonistic were defined in general. How did you determine which categories the list of behaviors was grouped in? 

One thing I would like to see addressed is why more hours of observation were not collected? 

Your pilot and observer training process was described well and was well thought out. 

In your paragraph on dominance hierarchies in the Discussion section, you only cite one study when talking about how the usefulness of the concept has been questioned. In order to validate this statement, more studies need to be cited here. 

When mentioning, "high levels of agonistic interactions directed at specific individuals , as we observed here directed towards A1 by A3 . . . " (lines 407 - 408 in the Discussion section), how did you determine levels were high? What measure is this being measured against? What is actually high in the wild vs. captivity? 

One general comment I have for the Discussion section is to be a little more cautious in your interpretations (e.g. mentioning A1 and A3 were not compatible due to their agonistic interactions - I don't think this can be determined with such a small data set in terms of hours of observations and with such a small sample size - you may be right but I would suggest being more cautious with the wording). 

Author Response

Thank you for your time on this review, your comments are appreciated. Responses to each comment can be found below.

I do think this manuscript is well written and that this study is a necessary study. The following are my comments on the manuscript:

In the Introduction section, you mention in the first paragraph that, "Agonistic interactions between elephants include threat displays, mock charges, biting, kicking, and other contact-based attacks that inflict pain . . ." My concern is how was it determined that these behaviors inflict pain? 

Excellent point, I have added the word ‘potentially’ to this section

There are a couple of citations missing, like Makecha et al. (2012), which concerns social relationships in captivity, and Garai (1992), which concerns special relationships in elephants (you  mention in the 3rd paragraph that it is not known if unrelated elephants brought together as adults in captivity can form such bonds. 

Thank you for these, it is frustrating that our literature search did not highlight them. I have added reference to both Makecha et al (2012) and Garai (1992), plus two others I found that cited the Garai (1992) study (Gobush & Wasser, 2009; and Bonaparte-Saller & Mench, 2018). I have also edited that section to say this now instead:

“Close associations between non-related individuals have been observed when elephants are raised together when populations are disturbed [19,20], as well as in captive adult female Asian elephants [21], but these likely act as substitute bonds in the absence of matrilineal kin [19] and may not be of equal quality to familial bonds”.

There are more captive studies on elephant social behavior than what is cited in the Introduction section (top of 4th paragraph), so I disagree with the statement starting with, "The few existing studies of captive Asian elephant social behavior . . . "

This was poorly worded, we meant to say reference the few studies of captive Asian elephant social behaviour that have included social data on related and unrelated individuals. Whilst other social behaviour studies have been done, it is not always possible from the data presented to dig down into related versus unrelated individuals. Unfortunately, in the period since I received your comments I have been unable to access a complete copy of Garai (1992) to check whether these data are in there also, so I have had to leave that study out from this section, but have referenced it elsewhere based upon the information in the abstract and added in reference here to Makecha et al (2012).

There are some spelling errors (not many) throughout the manuscript which can be easily addressed. One of the recurring error is using "zoo's" vs. "zoos"

These have been corrected, thank you

One of the things that I would like to see clarified are how affiliative and agonistic were defined in general. How did you determine which categories the list of behaviors was grouped in? 

These were based on an ethogram developed for the Elephant Welfare Group (including the affiliative and atomistic level groupings), which was based upon extensive literature review and expert consultation.

One thing I would like to see addressed is why more hours of observation were not collected? 

The video footage was limited to 4 days’ worth per time period, per zoo, between the hours of 9-5, as it was originally collected for other purposes. Due to limitations in coverage, we were only able to use footage from the indoor enclosures and thus set the collection windows to the times the elephants were most likely to be found indoors. Some individuals had significantly more observations (up to 10 hours’ worth in one sample period as can be seen in S. Table 3) but to be comparable we had to reduce the observation data to a rate of 5hrs per sample period as we had at least 5 hours of observations for every elephant.

Your pilot and observer training process was described well and was well thought out. 

Thank you!

In your paragraph on dominance hierarchies in the Discussion section, you only cite one study when talking about how the usefulness of the concept has been questioned. In order to validate this statement, more studies need to be cited here. 

Reference to three further manuscripts which discuss the usefulness of the concept and its limitations have now been added to support this statement; two of these were new references to the manuscript and are now numbers 43 and 44.

When mentioning, "high levels of agonistic interactions directed at specific individuals, as we observed here directed towards A1 by A3 . . . " (lines 407 - 408 in the Discussion section), how did you determine levels were high? What measure is this being measured against? What is actually high in the wild vs. captivity? 

The number of agonistic interactions directed from A1 to A3 were high within this study. The median frequency combined across both zoos was 3 and the mean was 5.2. A1 directed 36 instances of agonistic interactions towards A3, which is 12 times the median in these two groups, and 2.7 times more than the second most frequent number of agonistic interactions directed from one elephant to another in either zoo. Some of this has now been added to this section to justify the use of the term ‘high levels’.

Please also see the response below which further justifies the use of the term “high-levels” as compared to other captive elephants of comparable demography.

One general comment I have for the Discussion section is to be a little more cautious in your interpretations (e.g. mentioning A1 and A3 were not compatible due to their agonistic interactions - I don't think this can be determined with such a small data set in terms of hours of observations and with such a small sample size - you may be right but I would suggest being more cautious with the wording). 

Very good point, thank you. Based upon the other reviewers comments a mini-metanalysis has now been done comparing the hourly dyadic rate of interactions seen here to that from two other studies (Makecha et al 2012, and Coleing 2009). The rate of agonistic interactions seen between A1 and A3 was considerably higher than that seen in any other observed dyad. On this basis, the new text in this section reads thus:

“A1 was observed directing 36 agonistic interactions towards A3; this is 12 times the median frequency of agonistic interactions observed across both zoos, and 2.7 times higher than the second largest agonistic tie in either zoo. When combined with other published data [29,32] the mean hourly rate of agonistic interactions seen between 36 dyads of unrelated female Asian elephants was 0.36 interactions per hour, whilst for A1 and A3 the hourly rate was 3.7, which is the highest of all 36 dyads (S. Table 5). The average rate of agonistic interactions in related dyads from this comparison was 0.25, which although lower than for unrelate individuals, was not hugely so. High levels of agonistic interactions directed at specific individuals, as we observed here directed towards A1 by A3, suggest that the two individuals were not compatible.”

Reviewer 2 Report

In this manuscript, the authors investigate the potential influence of the kinship structure among groups of elephants housed together in zoos on their social behaviour, with potential implications for the welfare of these animals. In general, the manuscript is well written, addresses a relevant hypothesis, and appears to have been carefully conducted with clear reporting. 

My main issue with the manuscript in its current form is the obvious limitation in sample size - the study is based on an N = 2. The authors are aware of this limitation, but still present strong conclusions and do not attempt to assess the potential effect of these limitations.

I think that you should consider more carefully which other factors might explain the differences in social behaviour between the two groups. The 'limitation' section in the discussion currently focuses on technical issues. For example, I noticed that in the group that had higher rates of affiliative and lower rates of aggressive interactions two of the individuals were juveniles (below breeding age) and that their interactions appear to have shaped some of the patterns? Could the difference between the two groups simply be due to the age structure, rather than the relatedness structure? Do the interactions among the two adults in the related group still differ from those of any two adults in the unrelated group? 

The framing of this study against the previously conducted studies appears strange - my suggestion would be for the authors to integrate this knowledge. The effective difference in sample size is negligible (versus line 95), and it appears that the previously available evidence was sufficient for the development of guidelines (line 91). Is this true? What evidence were the current guidelines based on? The authors present their study as exploratory (we do not know enough about this subject) but it appears to rather be confirmatory (there is some evidence but we need more). This is simply about the framing - the study does not change at all in its value if it is confirmatory. However, it should be portrayed as such. 

More practically, would there be a way to integrate the previous results? Could you extract some statistical/other information from the earlier studies and present a quantitative synthesis in the results or the discussion (i.e. for each study list the rate of aggression in groups with related individuals versus groups with unrelated individuals)? This might also help to identify whether relatedness is of more importance as for example the age structure.

As I mentioned above, this is a carefully conducted study, and I am aware of the difficulties in obtaining these kinds of observations, so I think that there is sufficient material here to warrant publication. My comments are about the framing of this study, the acknowledgement of the limitations, the integration with the previous literature, and potential ways to expand by adding data already existing in the scientific literature.

Author Response

Comments and Suggestions for Authors

In this manuscript, the authors investigate the potential influence of the kinship structure among groups of elephants housed together in zoos on their social behaviour, with potential implications for the welfare of these animals. In general, the manuscript is well written, addresses a relevant hypothesis, and appears to have been carefully conducted with clear reporting. 

My main issue with the manuscript in its current form is the obvious limitation in sample size - the study is based on an N = 2. The authors are aware of this limitation, but still present strong conclusions and do not attempt to assess the potential effect of these limitations.

We hope that the corrections made in response to your specific comments below have addressed these concerns.

I think that you should consider more carefully which other factors might explain the differences in social behaviour between the two groups. The 'limitation' section in the discussion currently focuses on technical issues. For example, I noticed that in the group that had higher rates of affiliative and lower rates of aggressive interactions two of the individuals were juveniles (below breeding age) and that their interactions appear to have shaped some of the patterns? Could the difference between the two groups simply be due to the age structure, rather than the relatedness structure? Do the interactions among the two adults in the related group still differ from those of any two adults in the unrelated group? 

Agreed, this limitation could be made more clearly. Looking at the behaviour frequencies, only the affiliative network seems to have been impacted by the age structure, as the juveniles in Zoo B were responsible for the majority of the affiliative interactions seen there. However, in terms of NCD and agonist interactions, no clear age-related patterns can be seen. Further discussion of this has now been added. Further amendments are made in relation to this and are detailed in reply to the comment below on integrating the results.

The framing of this study against the previously conducted studies appears strange - my suggestion would be for the authors to integrate this knowledge. The effective difference in sample size is negligible (versus line 95), and it appears that the previously available evidence was sufficient for the development of guidelines (line 91). Is this true? What evidence were the current guidelines based on?

I believe the current BIAZA guidelines are based upon the status of wild elephants, rather than any specific evidence from captive elephants. They specifically state:

“Given that welfare is both difficult to define and measure we must accept that definitive welfare measures for zoos elephants will be difficult to attain, particularly given the number of elephants in captivity. In the absence of sufficient data we are obliged to give the animals the benefit of the doubt in terms of management recommendations. That is to say rather than waiting for evidence which states that a variable correlates welfare, where common sense suggests that such a variable is likely to correlate with welfare, we will assume it does until evidence is available to contradict this and management recommendations should reflect this.”

The authors present their study as exploratory (we do not know enough about this subject) but it appears to rather be confirmatory (there is some evidence but we need more). This is simply about the framing - the study does not change at all in its value if it is confirmatory. However, it should be portrayed as such. 

A few changes have been made to wording at the end of the Introduction around the aims to try and change the framing. I’m not entirely sure what more needs to be changed to alter how the study is portrayed but have also added this sentence to the limitations section:

As with other studies, the sample size here was small containing elephants from only two zoos. However, when integrated with other existing data, our results lend support to recommendations that relatedness is important when considering the welfare of captive elephants.”

More practically, would there be a way to integrate the previous results? Could you extract some statistical/other information from the earlier studies and present a quantitative synthesis in the results or the discussion (i.e. for each study list the rate of aggression in groups with related individuals versus groups with unrelated individuals)? This might also help to identify whether relatedness is of more importance as for example the age structure.

In an effort to integrate the results with those of other studies as suggested, two other papers were identified that provided frequency or rate of interaction data per dyad for captive female Asian elephants: Makecha et al (2012) and Coleing (2009). From these, hourly rates of affiliative and agonistic interactions have been extracted for each female-female dyad and combined with the data from the current study. This data has been shown in S. Tables 5 and 6. As you rightly suggested, this has helped to address the age-structure question, on which the following text has been added to the discussion:

“It could be argued that the higher rate of affiliative interactions observed amongst the related group is due to the younger age of B3 and B4 at Zoo B. However, when our data are combined with published data on the rate of interactions seen between female Asian elephants at two other zoos [29,32] the average hourly rate of affiliative interactions amongst 36 unrelated dyads was 0.98, and amongst 11 related dyads was 2.80 (S. Table 6). When interactions involving juveniles were removed, the average rate for related adult only individuals was still almost double that for unrelated individuals at 1.81, compared to 1.05. This lends strength to the supposition that the increased rate of affiliative interactions observed at zoo B was at least partly due to the relatedness of the group, and not the differential age-structure.”

This integration further enabled me to support the use of “high levels of agonistic interaction” when describing A1 and A3, about which the following text has been added:

“When combined with other published data [29,32] the mean hourly rate of agonistic interactions seen between 36 dyads of unrelated female Asian elephants was 0.36 interactions per hour, whilst for A1 and A3 the hourly rate was 3.7, which is the highest of all 36 dyads (S. Table 5). The average rate of agonistic interactions in related dyads from this comparison was 0.25, which although lower than for unrelate individuals, was not hugely so.”

As I mentioned above, this is a carefully conducted study, and I am aware of the difficulties in obtaining these kinds of observations, so I think that there is sufficient material here to warrant publication. My comments are about the framing of this study, the acknowledgement of the limitations, the integration with the previous literature, and potential ways to expand by adding data already existing in the scientific literature.

Thank you for your time on this review, your comments are appreciated.